# Sparse space-time deconvolution
# for Calcium image analysis

**Ferran Diego**     **Fred A. Hamprecht**
Heidelberg Collaboratory for Image Processing (HCI)
Interdisciplinary Center for Scientific Computing (IWR)
University of Heidelberg, Heidelberg 69115, Germany
`{ferran.diego,fred.hamprecht}@iwr.uni-heidelberg.de`

## Abstract

We describe a unified formulation and algorithm to find an extremely sparse representation for Calcium image sequences in terms of cell locations, cell shapes, spike timings and impulse responses. Solution of a single optimization problem yields cell segmentations and activity estimates that are on par with the state of the art, without the need for heuristic pre- or postprocessing. Experiments on real and synthetic data demonstrate the viability of the proposed method.

## 1   Introduction

A detailed understanding of brain function is a still-elusive grand challenge. Experimental evidence is collected mainly by electrophysiology and "Calcium imaging". In the former, multi-electrode array recordings allow the detailed study of hundreds neurons, while field potentials reveal the collective action of dozens or hundreds of neurons. The more recent Calcium imaging, on the other hand, is a fluorescent microscopy technique that allows the concurrent monitoring of the individual actions of thousands of neurons at the same time. While its temporal resolution is limited by the chemistry of the employed fluorescent markers, its great information content makes Calcium imaging an experimental technique of first importance in the study of neural processing, both *in vitro* [16, 6] and *in vivo* [5, 7]. However, the acquired image sequences are large, and in laboratory practice the analysis remains a semi-manual, tedious and subjective task.

Calcium image sequences reveal the activity of neural tissue over time. Whenever a neuron fires, its fluorescence signal first increases and then decays in a characteristic time course. Evolutionary and energetic constraints on the brain guarantee that, in most cases, neural activity is sparse in both space (only a fraction of neurons fire at a given instant) and time (most neurons fire only intermittently). The problem setting can be formalized as follows: given an image sequence as input, the desired output is (i) a set of cells[1] and (ii) a set of time points at which these cells were triggered. We here propose an unsupervised learning formulation and algorithm that leverages the known structure of the data to produce the sparsest representations published to date, and allow for meaningful automated analysis.

### 1.1   Prior Art

Standard laboratory practice is to delineate each cell manually by a polygon, and then integrate their fluorescence response over the polygon, for each point in time. The result is a set of time series, one per cell.

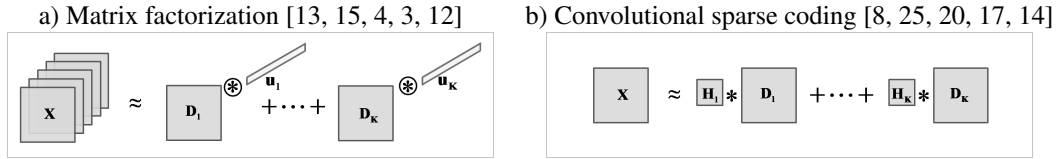

a) Matrix factorization [13, 15, 4, 3, 12]    b) Convolutional sparse coding [8, 25, 20, 17, 14]

Figure 1: Sketch of selected previous work. Left: Decomposition of an image sequence into a sum of $K$ components. Each component is given by the Cartesian product of a spatial component or basis image $\mathbf{D}_k$ and its temporal evolution $\mathbf{u}_k$. In this article, we represent such Cartesian products by the convolution of multidimensional arrays. Right: Description of a single image in terms of a sum of latent feature maps $\mathbf{D}_k$ convolved with filters $\mathbf{H}_k$

Given that the fluorescence signal impulse response to a stimulus is stereotypic, these time series can then be deconvolved to obtain a sparse temporal representation for each cell using nonnegative sparse deconvolution [24, 5, 10].

The problem of automatically identifying the cells has received much less attention, possibly due to the following difficulties [16, 23]: i) low signal-to-noise ratio (SNR); ii) large variation in luminance and contrast; iii) heterogeneous background; iv) partial occlusion and v) pulsations due to heartbeat or breathing in live animals. Existing work either hard-codes prior knowledge on the appearance of specific cell types [16, 22] or uses supervised learning (pixel and object level classification, [23]) or unsupervised learning (convolutional sparse block coding, [14]).

Closest in spirit to our work are attempts to simultaneously segment the cells *and* estimate their time courses. This is accomplished by matrix factorization techniques such as independent component analysis [13], nonnegative matrix factorization [12] and (hierarchical) dictionary learning [4, 3]. However, none of the above give results that are truly sparse in time; and all of the above have to go to some lengths to obtain reasonable cell segmentations: [13, 4] recur to heuristic post-processing, while [3] invokes structured sparsity inducing norms and [15] use an additional clustering step as initialization. All these extra steps are necessary to assure that each spatial component represents exactly one cell.

In terms of mathematical modeling, we build on recent advances and experiments in convolutional sparse coding such as [8, 25, 20, 17, 14]. Ref. [21] already applies convolutional sparse coding to video, but achieves sparsity only in space and not in time (where only pairs of frames are used to learn latent representations). Refs. [19, 18] consider time series which they deconvolve, without however using (or indeed needing, given their data) a sparse spatial representation.

## 1.2  Contributions

Summarizing prior work, we see three strands: i) Fully automated methods that require an extrinsic cell segmentation, but can find a truly[2] sparse representation of the temporal activity. ii) Fully automated methods that can detect and segment cells, but do not estimate time courses in the same framework. iii) Techniques that both segment cells and estimate their time courses. Unfortunately, existing techniques either produce temporal representations that are not truly sparse [12, 4, 3] or do not offer a unified formulation of segmentation and activity detection that succeeds without extraneous clustering steps [15].

In response, we offer the first unified formulation in terms of a single optimization problem: its solution simultaneously yields all cells along with their actions over time. The representation of activity is truly sparse, ideally in terms of a single nonzero coefficient for each distinct action of a cell. This is accomplished by sparse space-time deconvolution. Given a motion-corrected sequence of Calcium images, it estimates i) locations of cells and ii) their activity along with iii) typical cell shapes and iv) typical impulse responses. Taken together, these ingredients afford the sparsest, and thus hopefully most interpretable, representation of the raw data. In addition, our joint formulation

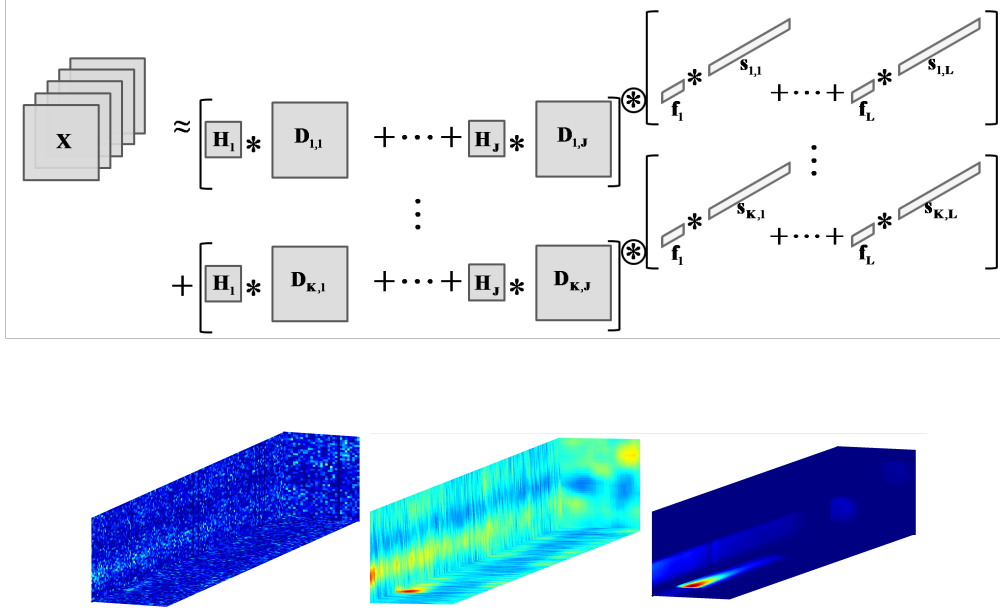

Figure 2: Summary of sparse space-time deconvolution. *Top*: Unified formulation in terms of a single optimization problem. *Bottom*: Illustration on tiny subset of data. Left: raw data. The fluorescence level to be estimated is heavily degraded by Poisson shot noise that is unavoidable at the requisite short exposure times. Middle: smoothed raw data. Right: approximation of the data in terms of a Cartesian product of estimated cell shapes and temporal activities. Each temporal activity is further decomposed as a convolution of estimated impulse responses and very few nonzero coefficients.

allows to estimate a nonuniform and temporally variable background. Experiments on difficult artificial and real-world data show the viability of the proposed formulation.

**Notation** Boldface symbols describe multidimensional arrays. We define $\mathbf{A} * \mathbf{B}$ as a convolution of multidimensional arrays $\mathbf{A}$ and $\mathrm{mirror}(\mathbf{B})$, with the result trimmed to the dimensions of $\mathbf{A}$. Here, the "mirror" operation flips a multidimensional array along every dimension. $\mathbf{A} \circledast \mathbf{B}$ is the full convolution result of multidimensional arrays $\mathbf{A}$ and $\mathrm{mirror}(\mathbf{B})$. These definitions are analogous to the "convn" command in matlab with shape arguments "same" and "full", respectively. $\| \cdot \|_0$ counts the number of non-zero coeficients, and $\| \cdot \|_F$ is the Frobenius norm.

## 2 Sparse space-time deconvolution (SSTD)

### 2.1 No background subtraction

An illustration of the proposed formulation is given in Fig. 2 and our notation is summarized in Table. 1. We seek to explain image sequence $\mathbf{X}$ in terms of up to $K$ cells and their activity over time. In so doing, all cells are assumed to have exactly one (Eq. 1.1) of $J << K$ possible appearances, and to reside at a unique location (Eq. 1.1). These cell locations are encoded in $K$ latent binary feature maps. The activity of each cell is further decomposed in terms of a convolution of impulses (giving the precise onset of each burst) with exactly one of $L << K$ types of impulse responses. A single cell may "use" different impulse responses at different times, but just one type at any one time ((Eq. 1.2).

All of the above is achieved by solving the following optimization problem:

$$\min_{\mathbf{D},\mathbf{H},\mathbf{f},\mathbf{s}} \left\| \mathbf{X} - \sum_{k=1}^{K} \left( \sum_{j=1}^{J} \mathbf{D}_{k,j} * \mathbf{H}_j \right) \circledast \left( \sum_{l=1}^{L} \mathbf{s}_{k,l} * \mathbf{f}_l \right) \right\|_F^2 \qquad (1)$$

| Constraint | | Interpretation |
|---|---|---|
| $\sum_j \|\mathbf{D}_{k,j}\|_0 \leq 1, \forall k$ | (1.1) | at most one location and appearance per component |
| $\sum_l \|s_{t,k,l}\|_0 \leq 1, \forall k,t$ | (1.2) | only one type of activation at each time per cell |
| $\|\mathbf{H}_j\|_F^2 \leq 1, \forall j$ | (1.3) | prevent cell appearance from becoming large |
| $\|\mathbf{f}_l\|_2^2 \leq 1, \forall l$ | (1.4) | prevent impulse filter from becoming large |

(The "such that" label appears to the left of constraints (1.1)–(1.4).)

Here, the optimization is with respect to the cell detection maps $\mathbf{D}$, cell appearances $\mathbf{H}$, activity patterns or impulse responses $\mathbf{f}$ as well as "truly sparse" activity indicator vectors $\mathbf{s}$. This optimization is subject to the two constraints mentioned earlier plus upper bounds on the norm of the learned filters.

The user needs to select the following parameters: an upper bound $K$ on the number of cells as well as the size in pixels $H$ of their matched filters / convolution kernels $\mathbf{H}$; upper bounds $J$ on the number of different appearances and $L$ on the number of different activity patterns that cells may have; as well as the number of coefficients $F$ that the learned impulse responses may have. Considering that we propose a method for both cell detection and sparse time course estimation, the number of six user-adjustable parameters compares favourably to previous work. Methods that decouple these steps typically need more parameters altogether, and the heuristics that prior work on joint optimization uses also have a large number of (implicit) parameters.

While many other approximations such as $\sum_{k=1}^K \mathbf{D}_k \circledast \mathbf{s}_k * \mathbf{f}_k$ or $\sum_{k=1}^K \sum_{j=1}^J \mathbf{D}_{k,j} * \mathbf{H}_j \circledast \mathbf{s}_{k,j} * \mathbf{f}_j$ are conceivable and may make sense in other applications areas, the proposed formulation is the most parsimonious of its kind. Indeed, it uses a small pool of $J$ shapes and $L$ firing patterns, which can be combined freely, to represent *all* cells and their activities. It is owing to this fact that we dub the method sparse spatio-temporal deconvolution (SSTD).

## 2.2 SSTD with background subtraction

In actual experiments, the observed fluorescence level is a sum of the signal of interest plus a nuisance background signal. This background is typically nonuniform in the spatial domain and, while it can be modeled as constant over time [15, 24], is often also observed to vary over time, prompting robust local normalization as a preprocessing step [7, 4].

Here, we generalize the formulation from (1) to correct for a background that is assumed to be spatially smooth and time-varying. In more detail, we model the background in terms of the direct product $\mathbf{B}^s \circledast \mathbf{b}^t$ of a spatial component $\mathbf{B}^s \in \mathbb{R}_+^{M \times N \times 1}$ and a time series $\mathbf{b}^t \in \mathbb{R}_+^{1 \times 1 \times T}$. Insights into the physics and biology of Calcium imaging suggest that (except for saturation regimes characterized by high neuron firing rates), it is reasonable to assume that the normalized quantity (observed fluorescence minus background) divided by background, typically dubbed $\Delta\mathsf{F}/\mathsf{F}_0$, is linearly related to the intracellular Calcium concentration [24, 10]. In keeping with this notion, we now propose our final model, viz.

$$\min_{\mathbf{D},\mathbf{H},\mathbf{f},\mathbf{s},\mathbf{B}^s,\mathbf{b}^t} \left\| \left( \mathbf{X} - \sum_{k=1}^K \left( \sum_{j=1}^J \mathbf{D}_{k,j} * \mathbf{H}_j \right) \circledast \left( \sum_{l=1}^L \mathbf{s}_{k,l} * \mathbf{f}_l \right) - \mathbf{B}^s \circledast \mathbf{b}^t \right) \oslash \left( \mathbf{B}^s \circledast \mathbf{b}^t \right) \right\|_F^2$$

$$+ \lambda \|\mathbf{B}^s\|_{TV} \text{ such that } (1.1) - (1.4), \ \mathbf{B}^s > 0, \ \mathbf{b}^t > 0 \tag{2}$$

with "$\oslash$" denoting an elementwise division. Note that the optimization now also runs over the spatial and temporal components of the background, with the total variation (TV) regularization term[3] enforcing spatial smoothness of the spatial background component [2].

In addition to the previously defined parameters, the user also needs to select parameter $\lambda$ which determines the smoothness of the background estimate.

## 2.3 Optimization

The optimization problem in (2) is convex in either the spatial or the temporal filters $\mathbf{H}$, $\mathbf{f}$ alone when keeping all other unknowns fixed; but it is nonconvex in general. In our experiments, we use a block coordinate descent strategy [1, Section 2.7] that iteratively optimizes one group of variables while

| Symbol | Definition |
|---|---|
| $\mathbf{X} \in \mathbb{R}_+^{M \times N \times T}$ | image sequence of length $T$, each image is $M \times N$ |
| $K \in \mathbb{N}_+$ | number of cells |
| $J \in \mathbb{N}_+$ | number of distinct cell appearances |
| $\mathbf{H}_j \in \mathbb{R}_+^{H \times H \times 1}$ | $j^{th}$ cell appearance / spatial filter / matched filter of size $H \times H$ |
| $\mathbf{D}_{k,j} \in \{0,1\}^{M \times N \times 1}$ | indicator matrix of the $k^{th}$ cell for the $j^{th}$ cell appearance |
| $L \in \mathbb{N}_+$ | number of distinct impulse responses / activity patterns |
| $\mathbf{f}_l \in \mathbb{R}_+^{1 \times 1 \times F}$ | $l^{th}$ impulse response of length $F$ |
| $\mathbf{s}_{k,l} \in \mathbb{R}_+^{1 \times 1 \times T}$ | indicator vector of the $k^{th}$ spike train for the $l^{th}$ impulse response |

Table 1: Notation

fixing all others (see supplementary material for details). The nonconvex $l_0$-norm constraints require that cell centroids $\mathbf{D}$ and spike trains $\mathbf{s}$ are estimated by techniques such as convolutional matching pursuit [20]; while the spatio-temporal filters can be learned using simpler gradient descent [25], K-SVD [20] or simple algebraic expressions.

All unknowns are initialized with standard Gaussian noise truncated to nonnegative values. The limiting number of cells $K$ can be set to a generous upper bound of the expected true number because spatial components without activity are automatically set to zero during optimization.

## 3 Experimental Setup

This section describes the data and algorithms used for experiments and benchmarks.

### 3.1 Inferring Spike Trains

The following methods assume that cell segmentation has already been performed by some means, and that the fluorescence signal of individual pixels has been summed up for each cell and every time step. They can hence concentrate exclusively on the estimation of a "truly sparse" representation of the respective activities in terms of a "spike train".

**Data** We follow [24, 5] in generating 1100 sequences consisting of one-sided exponential decays with a constant amplitude of 1 and decay rate $\tau = 1/2s$, sampled at $30fps$ with firing rates ranging uniformly from 1 to $10Hz$ and different Gaussian noise levels $\sigma \in [0.1, 0.6]$.

**Fast non-negative deconvolution (FAST)** [24] uses a one-sided exponential decay as parametric model for the impulse response by invoking a first-order autoregressive process. Given that our artificial data is free of a nuisance background signal, we disregard its ability to also model such background. The sole remaining parameter, the rate of the exponential decay, can be fit using maximum likelihood estimation or a method-of-moments approach [15].

**Peeling** [5] finds spikes by means of a greedy approach that iteratively removes one impulse response at a time from the residual fluorescence signal. Importantly, this stereotypical transient must be manually defined *a priori*.

**Sparse temporal deconvolution (STD)** with a single impulse response is a special case of this work for given nonoverlapping cell segmentations and $L = 1$; and it is also a special case of [14]. The impulse response can be specified beforehand (amounting to sparse coding), or learned from the data (that is, performing dictionary learning on time-series data).

### 3.2 Segmenting Cells *and* Estimating Activities

**Data** Following the procedure described in [4, 12, 13], we have created 80 synthetic sequences with a duration of $15s$ each at a frame rate of $30fps$ with image sizes $M = N = 512$ pixels. The cells are randomly selected from 36 cell shapes extracted from real data, and are randomly located in different locations with a maximum spatial overlap of $30\%$. Each cell fires according to a dependent Poisson process, and its activation pattern follows a one-sided exponential decay with

a scale selected uniform randomly between $500$ and $800ms$. The average number of active cells per frame varies from 1 to 10. Finally, the data has been distorted by additive white Gaussian noise with a relative amplitude (max. intensity $-$ mean intensity)$/\sigma_{noise} \in \{3, 5, 7, 10, 12, 15, 17, 20\}$. By construction, the identity, location and activity patterns of all cells are known. The supplemental material shows an example with its corresponding inferred neural activity.

Real-world data comes from two-photon microscopy of mouse motor cortex recorded in vivo [7] which has been motion-corrected. These sequences allow us to conduct qualitative experiments.

**ADINA** [4] relies on dictionary learning [11] to find both spatial components and their time courses. Both have many zero coefficients, but are not "truly sparse" in the sense of this paper. The method comes with a heuristic post-processing to separate coactivated cells into distinct spatial components.

**NMF+ADINA** uses non-negative matrix factorization to infer both the spatial and temporal primitives of an image sequence as in [12, 15]. In contrast to [15] which uses a k-means clustering of highly confident spike vectors to provide a good initialization in the search for spatial components, we couple NMF with the postprocessing of ADINA.

**CSBC+SC** combines convolutional sparse block coding [14] based on a single still image (obtained from the temporal mean or median image, or a maximum intensity projection across time) with temporal sparse coding.

**CSBC+STD** combines convolutional sparse block coding [14] based on a single still image (obtained from the temporal mean or median image, or a maximum intensity projection across time) with the proposed sparse temporal deconvolution in Sect. 3.1.

**SSTD** is the method described here. We used $J = L = 2, K = 200, F = 200$ and $H = 31, 15$ for the artificial and real data, respectively.

## 4 Results

### 4.1 Inferring spike trains

To quantify the accuracy of activity detection, we first threshold the estimated activities and then compute, by summing over each step in every time series, the number of true and false negatives and positives. For a fair comparison, the thresholds were adjusted separately for each method to give optimal accuracy. Sensitivity, precision and accuracy computed from the above implicitly measure both the quality of the segmentation and the quality of the activity estimation. An additional measure, SPIKE distance [9], emphasizes any temporal deviations between the true and estimated spike location in a truly sparse representation.

Fig. 3 shows that, unsurprisingly, best results are obtained when methods use the true impulse response rather than learning it from the data. This finding does not carry over to real data, where a "true" impulse response is typically not known. Given the true impulse response, both FAST and STD fare better than Peeling, showing that a greedy algorithm is faster but gives somewhat worse results. Even when learning the impulse response, FAST and STD are no worse than Peeling. When learning the parameters, FAST has an advantage over STD on this artificial data because FAST already uses the correct parametric form of the impulse response that was used to generate the data and only needs to learn a single parameter; while STD learns a more general but nonparametric activity model with many degrees of freedom.

The great spread of all quality measures results from the wide range of noise levels used, and the overall deficiencies in accuracy attest to the difficulty of these simulated data sets.

### 4.2 Segmenting Cells *and* Inferring spike trains

Fig. 4 shows that all the methods from Sect. 3.2 reach respectable and comparable performance in the task of identifying neural activity from non-trivial synthetic image sequences.

CSBC+SC reaches the highest sensitivity while SSTD has the greatest precision. SSTD apparently achieves comparable performance to the other methods without the need for a heuristic pre- or postprocessing. Multiple random initializations lead to similar learned filters (results not shown),

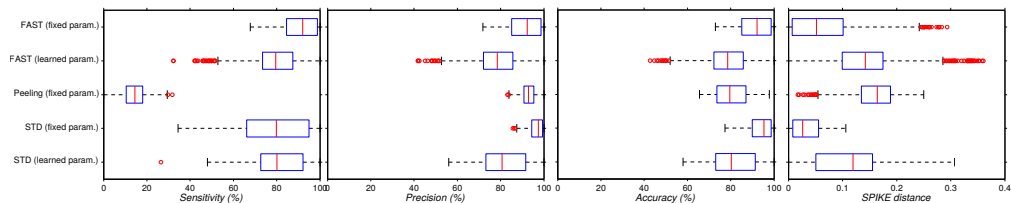

Figure 3: Sensitivity, precision, accuracy (higher is better) and SPIKE distance (lower is better) of different methods for spike train estimation. Methods that need to learn the activation pattern perform worse than those using the true (but generally unknown) activation pattern and its parameters. FAST is at an advantage here because it happens to use the very impulse response that was used in generating the data.

so the optimization problem seems to be well-posed. The price to pay for the elegance of a unified formulation is a much higher computational cost of this more involved optimization. Again, the spread of sensitivities, precisions and accuracies results from the range of noise levels used in the simulations. The plots suggest that SSTD may have fewer "catastrophic failure" cases, but an even larger set of sequences will be required to verify this tendency.

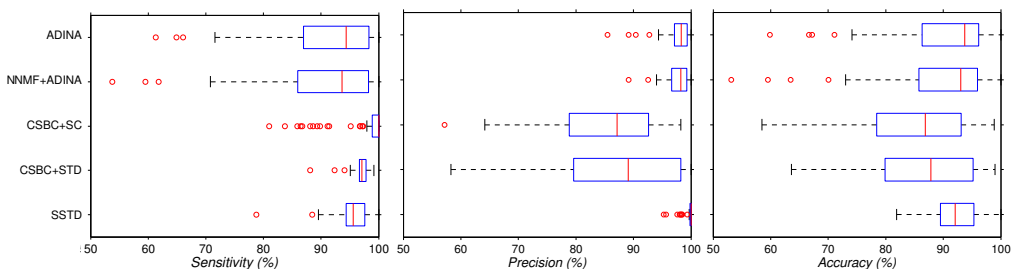

Figure 4: Quality of cell detection and and the estimation of their activities. SSTD does as well as the competing methods that rely on heuristic pre- or post-processing.

**Real Sequences:** Qualitative results are shown in Fig. 5. SSTD is able to distinguish both cells with spatial overlap and with high temporal correlation. It compensates large variations in luminance and contrast, and can discriminate between different types of cells. Exploiting truly sparse but independent representations in both the spatial and the temporal domain allows to infer plausible neural activity and, at the same time, reduce the noise in the underlying Calcium image sequence.

## 5  Discussion

The proposed SSTD combines the decomposition of the data into low-rank components with the finding of a convolutional sparse representation for each of those components. The formalism allows exploiting sparseness and the repetitive motifs that are so characteristic of biological data. Users need to choose the number and size of filters that indirectly determine the number of cell types found and their activation patterns.

As shown in Fig. 5, the approach gives credible interpretations of raw data in terms of an extremely sparse and hence parsimonious representation.

The decomposition of a spacetime volume into a Cartesian product of spatial shapes and their time courses is only possible when cells do not move over time. This assumption holds for in vitro experiments, and can often be satisfied by good fixation in in vivo experiments, but is not universally valid. Correcting for motions in a generalized unified framework is an interesting direction for future work. The experiments in section 4.1 suggest that it may also be worthwhile to investigate the use of more parametric forms for the impulse response instead of the completely unbiased variant used here.

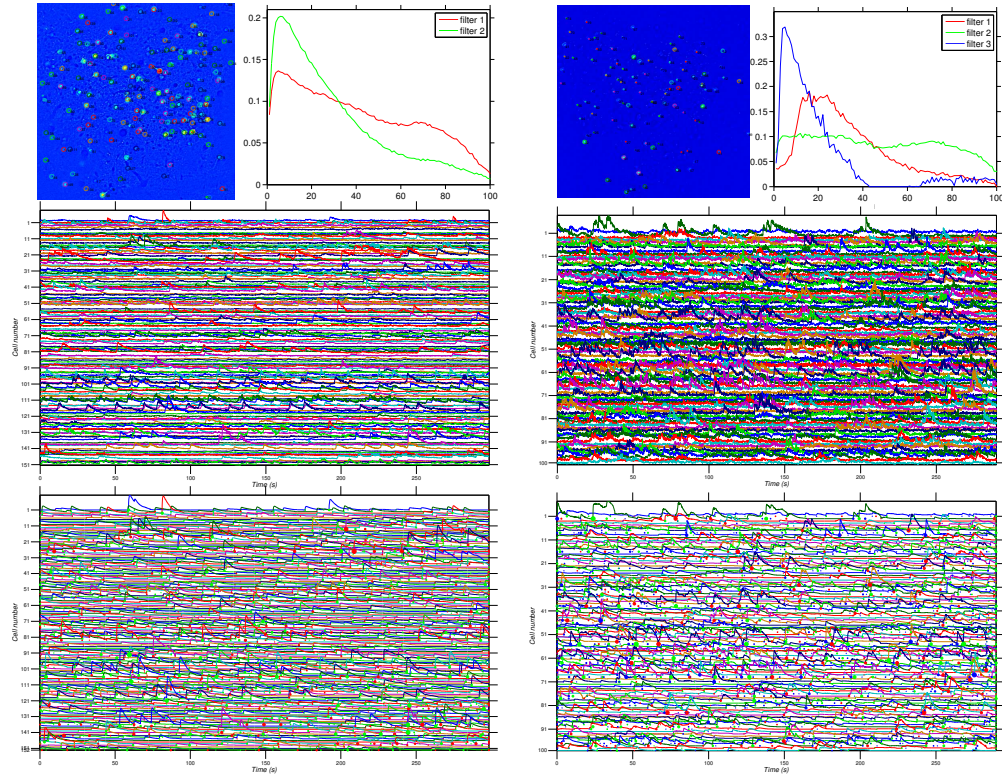

Figure 5: Qualitative results on two real data sets. The data on the left column shows mostly cell bodies, while the data on the right shows both cell bodies (large) and dendrites (small). For each data set, the top left shows an average projection of the relative fluorescence change across time with cell centroids $\mathbf{D}$ (black dots) and contours of segmented cells, and the top right shows the learned impulse responses. In the middle, the fluorescence levels integrated over the segmented cells are shown in random colors. The bottom shows by means of small disks the location, type and strength of the impulses that summarize all the data shown in the middle. Together with the cell shapes, the impulses from part of the "truly sparse" representation that we propose. When convolving these spikes with the impulse responses from the top right insets, we obtain the time courses shown in random colors.

Such advances will further help making Calcium imaging an enabling tool for the neurosciences.

## Footnotes

[1]Optical sectioning by techniques such as confocal or two-photon microscopy implies that we see only parts of a neuron, such as a slice through its cell body or a dendrite, in an image plane. For brevity, we simply refer to these as "cells" in the following.

[2]We distinguish a sparse representation, in which the estimated time course of a cell has many zeros; and a "truly sparse" representation in which a single action of a cell is ideally represented in terms of a single nonzero coefficient.

[3]TV measures the sum of the absolute values of the spatial gradient.

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
