[Supplementary Material]

# Supplementary material for the paper:
## *Sparse Space-Time Deconvolution for Calcium Image Analysis*

**Ferran Diego** **Fred A. Hamprecht**
Heidelberg Collaboratory for Image Processing (HCI)
Interdisciplinary Center for Scientific Computing (IWR)
University of Heidelberg, Heidelberg 69115, Germany
{ferran.diego,fred.hamprecht}@iwr.uni-heidelberg.de

## 3 Optimization

We now consider the optimization in Eq. (1) where $\bar{\mathbf{X}}, \bar{\mathbf{D}}, \bar{\mathbf{S}}, \bar{\mathbf{H}}$ and $\bar{\mathbf{f}}$ are the non-normalized variables w.r.t. the baseline concentration. The problem is not *jointly* convex, but becomes convex w.r.t. one variable while keeping fixed the others since the total variation regularization term is also convex. Hence, this problem is optimized using a block coordinate descent strategy [1, Section 2.7] that iteratively optimizes a group of variables while fixing the others. In addition. the $l_0$-norm constraints requires that cell centroids and spike train are estimated by convolutional matching pursuit [11]; while the spatio-temporal filters can be learnt using gradient descent [12], K-SVD approach [11], or closed-form formulation.

$$
\min_{\bar{\mathbf{D}},\bar{\mathbf{H}},\bar{\mathbf{f}},\bar{\mathbf{S}},\mathbf{B}^s,\mathbf{b}^t} \left\| \left( \bar{\mathbf{X}} - \sum_{k=1}^{K} \left( \sum_{j=1}^{J} \bar{\mathbf{D}}_{k,j} * \bar{\mathbf{H}}_j \right) \circledast \left( \sum_{l=1}^{L} \bar{\mathbf{s}}_{k,l} * \bar{\mathbf{f}}_l \right) - \mathbf{B}^s \circledast \mathbf{b}^t \right) \oslash \left( \mathbf{B}^s \circledast \mathbf{b}^t \right) \right\|_F^2
$$

$$
+ \lambda \|\mathbf{B}^s\|_{TV} \text{ s.t. } \begin{cases} \sum_{j=1}^{J} \|\bar{\mathbf{D}}_{k,j}\|_0 \leq 1, \forall k \\ \left\| \bar{\mathbf{H}}_j \right\|_F^2 \leq 1, \forall j \\ \sum_{l=1}^{L} \|\bar{s}_{t,k,l}\|_0 \leq 1, \forall k,t \\ \|\bar{\mathbf{f}}_l\|_2^2 \leq 1, \forall l \\ \bar{\mathbf{D}}_{k,j} \geq 0, \bar{\mathbf{H}}_j \geq 0, \bar{\mathbf{f}}_l \geq 0, \bar{\mathbf{s}}_{k,l} \geq 0, \forall k,j,l \\ \mathbf{B}^s > 0, \mathbf{b}^t > 0 \end{cases} \tag{1}
$$

The spatial vectorization of the full convolution between a matrix $\mathbf{D}$ and a vector $\mathbf{u}$ ( $\mathbf{D} \circledast \mathbf{u}$) satisfies the following equality: $\text{vec}(\mathbf{D} \circledast \mathbf{u}) = \text{vec}(D)\mathbf{u}^T$. Assuming $\mathbf{X} \in \mathbb{R}^{MN \times T}$ from now on, Eq. (1) is rewritten as:

$$
\min_{\bar{\mathbf{D}},\bar{\mathbf{H}},\bar{\mathbf{f}},\bar{\mathbf{S}},\mathbf{B}^s,\mathbf{b}^t} \left\| \left( \bar{\mathbf{X}} - \sum_{k=1}^{K} \text{vec} \left( \sum_{j=1}^{J} \bar{\mathbf{D}}_{k,j} * \bar{\mathbf{H}}_j \right) \left[ \sum_{l=1}^{L} \bar{\mathbf{s}}_{k,l} * \bar{\mathbf{f}}_l \right]^T - \text{vec}(\mathbf{B}^s)[\mathbf{b}^t]^T \right) \oslash \left( \text{vec}(\mathbf{B}^s)[\mathbf{b}^t]^T \right) \right\|_F^2
$$

$$
+ \lambda \|\mathbf{B}^s\|_{TV} \text{ s.t. } \begin{cases} \sum_{j=1}^{J} \|\bar{\mathbf{D}}_{k,j}\|_0 \leq 1, \forall k \\ \left\| \bar{\mathbf{H}}_j \right\|_F^2 \leq 1, \forall j \\ \sum_{l=1}^{L} \|\bar{s}_{t,k,l}\|_0 \leq 1, \forall k,t \\ \|\bar{\mathbf{f}}_l\|_2^2 \leq 1, \forall l \\ \bar{\mathbf{D}}_{k,j} \geq 0, \bar{\mathbf{H}}_j \geq 0, \bar{\mathbf{f}}_l \geq 0, \bar{\mathbf{s}}_{k,l} \geq 0, \forall k,j,l \\ \mathbf{B}^s > 0, \mathbf{b}^t > 0 \end{cases} \tag{2}
$$

## 3.1 Update of baseline Calcium Concentration

The spatial and temporal component of the baseline Calcium concentration $\{\text{vec}(\mathbf{B}^s), \mathbf{b}^t\}$ is approximated by the decomposition of the residual data $\tilde{\mathbf{X}}$ into two matrices of rank 1, and it is also used to normalized it into relative percentage of fluorescence change. This estimation is achieved by minimizing:

$$\min_{\mathbf{B}^s > \mathbf{0}, \mathbf{b}^t > \mathbf{0}} \left\| \left( \tilde{\mathbf{X}} - \text{vec}(\mathbf{B}^s)[\mathbf{b}^t]^T \right) \oslash \left( \text{vec}(\mathbf{B}^s)[\mathbf{b}^t]^T \right) \right\|_F^2 + \lambda \|\mathbf{B}^s\|_{TV}, \tag{3}$$

where $\tilde{\mathbf{X}} = \bar{\mathbf{X}} - \sum_{k=1}^K \text{vec} \left( \sum_{j=1}^J \bar{\mathbf{D}}_{k,j} * \bar{\mathbf{H}}_j \right) \left[ \sum_{l=1}^L \bar{\mathbf{s}}_{k,l} * \bar{\mathbf{f}}_l \right]^T$, and $\| \cdot \|_{TV}$ is any total variation regularization term as isotropic, anisotropic, high-order, and non-local.

Instead of minimizing directly Eq. (3), we infer both components as a standard non-negative matrix factorization with the constraint of being strictly positive. The estimated spatial and temporal component are then used to normalize the residual reconstruction. This approximation still computes relative fluorescence changes while avoids the preference of matching large intensities on Eq. (3).

Following [3, Lemma3.1], [7, 4] and the alternation between multiplicative update rules, the spatial component $\mathbf{B}^s$ is inferred by taking advantage of well-developed total-variation solvers on

$$\min_{\mathbf{B}^s > \mathbf{0}} \left\| \frac{\tilde{\mathbf{X}} \mathbf{b}^t}{\|\mathbf{b}^t\|_2^2} - \text{vec}(\mathbf{B}^s) \right\|_F^2 + \lambda \|\mathbf{B}^s\|_{TV}, \tag{4}$$

while the temporal component $\mathbf{b}^t$ is inferred by minimizing

$$\min_{\mathbf{b}^t > \mathbf{0}} \left\| \frac{\tilde{\mathbf{X}}^T \text{vec}(\mathbf{B}^s)}{\|\text{vec}(\mathbf{B}^s)\|_2^2} - \mathbf{b}^t \right\|_F^2. \tag{5}$$

## 3.2 Change of variables before updating sparse space-time deconvolution terms

The optimization of the spatio-temporal filters and its corresponding cell centroid and spike train is achieved using block-coordinate descent [1, Section 2.7] due to the joint non-convexity. For simplicity on the update formulas, we apply the next change of variables:

$$\mathbf{X} = \left( \bar{\mathbf{X}} - \text{vec}(\mathbf{B}^s)[\mathbf{b}^t]^T \right) \oslash \left( \text{vec}(\mathbf{B}^s)[\mathbf{b}^t]^T \right) \tag{6}$$

and

$$\sum_{k=1}^K \text{vec} \left( \sum_{j=1}^J \mathbf{D}_{k,j} * \mathbf{H}_j \right) \left[ \sum_{l=1}^L \mathbf{s}_{k,l} * \mathbf{f}_l \right]^T = \left( \sum_{k=1}^K \text{vec} \left( \sum_{j=1}^J \bar{\mathbf{D}}_{k,j} * \bar{\mathbf{H}}_j \right) \left[ \sum_{l=1}^L \bar{\mathbf{s}}_{k,l} * \bar{\mathbf{f}}_l \right]^T \right) \oslash \left( \text{vec}(\mathbf{B}^s)[\mathbf{b}^t]^T \right). \tag{7}$$

Hence, the minimization of Eq. (2) is rewritten as:

$$\min_{\mathbf{D}, \mathbf{H}, \mathbf{f}, \mathbf{S}} \left\| \mathbf{X} - \sum_{k=1}^K \text{vec} \left( \sum_{j=1}^J \mathbf{D}_{k,j} * \mathbf{H}_j \right) \left[ \sum_{l=1}^L \mathbf{s}_{k,l} * \mathbf{f}_l \right]^T \right\|_F^2 \quad \text{s.t.} \begin{cases} \sum_{j=1}^J \|\mathbf{D}_{k,j}\|_0 \leq 1, \forall k \\ \|\mathbf{H}_j\|_F^2 \leq 1, \forall j \\ \sum_{l=1}^L \|s_{t,k,l}\|_0 \leq 1, \forall k, t \\ \|\mathbf{f}_l\|_2^2 \leq 1, \forall l \\ \mathbf{D}_{k,j} \geq 0, \mathbf{H}_j \geq 0, \mathbf{f}_l \geq 0, \mathbf{s}_{k,l} \geq 0, \forall k, j, l \end{cases} \tag{8}$$

## 3.3 Update Cell Centroids $\mathbf{D}_{k,j}$

The optimization of $\mathbf{D}_{k,j}$ includes a $l_0$-norm that aims to infer, at most, one location of only one cell type per basis function. This minimization is written as

$$\min_{\mathbf{D}} \left\| \mathbf{X} - \sum_{k=1}^{K} \text{vec} \left( \sum_{j=1}^{J} \mathbf{D}_{k,j} * \mathbf{H}_j \right) \tilde{\mathbf{u}}_k^T \right\|_F^2 \text{ s.t. } \left\{ \begin{array}{l} \sum_{j=1}^{J} \|\mathbf{D}_{k,j}\|_0 \leq 1, \forall k \\ \mathbf{D}_{k,j} \geq 0, \forall k, j \end{array} \right. , \qquad (9)$$

where $\tilde{\mathbf{u}}_k = \sum_{l=1}^{L} \mathbf{s}_{k,l} * \mathbf{f}_l$ is the result of the temporal convolutional sparse coding of the $k^{th}$ basis function. The inference of Eq. (9) is NP-hard due to $l_0$-norm constraints, but greedy algorithms as matching pursuit [8] has been proposed to handle this problem and achieved good performance, *e.g.* patch-based dictionary learning [10]. Given a signal and a dictionary, this algorithm iteratively chooses a basis function on the dictionary that most reduces the reconstruction error given the current residual. In our case, we aim to find the basis function, the filter, and the cell location that most decreases the following reconstruction error without constraints

$$\min_{\mathbf{D} \geq 0} \left\| \mathbf{X} - \sum_{k=1}^{K} \text{vec} \left( \sum_{j=1}^{J} \mathbf{D}_{k,j} * \mathbf{H}_j \right) \tilde{\mathbf{u}}_k^T \right\|_F^2 =$$

$$= \min_{\mathbf{D} \geq 0} \text{tr} \left( \mathbf{X}^T \mathbf{X} - 2 \sum_{j,k} \tilde{\mathbf{u}}_k \mathbf{D}_{:,k,j}^T \tilde{\mathbf{H}}_j^T \mathbf{X} + \sum_{k,j,r,q} \tilde{\mathbf{u}}_k \mathbf{D}_{:,k,j}^T \tilde{\mathbf{H}}_j^T \tilde{\mathbf{H}}_q \mathbf{D}_{:,r,q} \tilde{\mathbf{u}}_r^T \right), \qquad (10)$$

where $\mathbf{D}_{:,k,j}$ is the vectorization of the $k^{th}$ basis function of the $j^{th}$ filter $\mathbf{D}_{k,j}$, $\text{tr}(\cdot)$ is the trace operator, and $\tilde{\mathbf{H}}_j$ is the Toeplitz matrix of the $j^{th}$ filter. The convolutional matching pursuit for inferring cell location and type is summarized in Algorithm 1. Step 5 can be accelerated by applying the Gramm matrix on the Toeplitz matrix at each time a basis function is updated as done in [11, 9, 8]. In contrast, step 11 projects cell centroids to the convex set defined in the main manuscript.

---

**Algorithm 1** Convolutional Matching Pursuit for inferring $\mathbf{D}_{k,j}$

---

1: **Input:** Data matrix $\mathbf{X}$, number of basis functions $K$ and of spatial filters $J$, filters $\mathbf{H}_j$ and temporal descriptions $\tilde{\mathbf{U}} = [\tilde{\mathbf{u}}_1 \ldots \tilde{\mathbf{u}}_K]$.
2: **Initialization:** $\mathbf{D}_{:,k,j} = \mathbf{0} \; \forall k, j$, and active set $\Lambda = \{1, \ldots, K\}$
3: **for** $k = 1, \ldots, K$ **do**
4:     **for** $i \in \Lambda$ and $j = 1, \ldots, J$ **do**
5:         **Update residual gradient**: $\dfrac{\partial}{\partial \mathbf{D}_{:,i,j}} := \tilde{\mathbf{H}}_j^T \mathbf{X} \tilde{\mathbf{u}}_i - \sum_{q,r} \tilde{\mathbf{u}}_q^T \tilde{\mathbf{u}}_i \tilde{\mathbf{H}}_j^T \tilde{\mathbf{H}}_r \mathbf{D}_{:,q,r},$
6:     **end for**
7:     **Find** $[c^\star, i^\star, j^\star] = \arg\max_{c, i \in \Lambda, j} \dfrac{\partial}{\partial \mathbf{D}_{c,i,j}}$
8:     **Set** $D_{c^\star, i^\star, j^\star} = \dfrac{\partial}{\partial \mathbf{D}_{c^\star, i^\star, j^\star}}$
9:     **Remove from active set** $\Lambda = \Lambda - \{i^\star\}$
10: **end for**
11: **Set** $\mathbf{D}_{:,q,r} = \text{sign}(\mathbf{D}_{:,q,r})$
12: **Output:** Updated the location and type of cell per each basis function $\mathbf{D}_{k,j}$.

---

## 3.4 Update Spike Train $\mathbf{s}_{k,l}$

The optimization of the spike train $\mathbf{s}_{k,l}$ also includes a $l_0$-norm that avoids the use of multiple activation patterns at the same time in the same basis function. Then, the spike train is inferred by minimizing

$$\min_{\mathbf{S} \geq 0} \left\| \mathbf{X} - \sum_{k=1}^{K} \tilde{\mathbf{d}}_k \left[ \sum_{l=1}^{L} \mathbf{s}_{k,l} * \mathbf{f}_l \right]^T \right\|_F^2 \quad \text{s.t.} \quad \sum_{l=1}^{L} \|s_{t,k,l}\|_0 \leq 1, \forall k, t, \tag{11}$$

where $\tilde{\mathbf{d}}_k = \sum_{j=1}^{J} \mathbf{D}_{k,j} * \mathbf{H}_j$ is the vectorized description of the $k^{th}$ cell (basis function) and $\mathbf{S}$ is the concatenation of all the spike trains $\mathbf{S} = [\mathbf{s}_{1,1} \ldots \mathbf{s}_{1,L} \ldots \mathbf{s}_{K,1} \ldots \mathbf{s}_{K,L}]$. Following the similarities of the previous section, we again use convolutional matching pursuit to infer the locations and the activation patterns that most decreases the current reconstruction error without constraints

$$\min_{\mathbf{S} \geq 0} \left\| \mathbf{X} - \sum_{k=1}^{K} \tilde{\mathbf{d}}_k \left[ \sum_{l=1}^{L} \mathbf{s}_{k,l} * \mathbf{f}_l \right]^T \right\|_F^2$$

$$= \min_{\mathbf{S} \geq 0} \text{tr} \left( \mathbf{X}^T \mathbf{X} - 2 \sum_{k,l} \tilde{\mathbf{F}}_l^T \mathbf{X}^T \tilde{\mathbf{d}}_k \mathbf{s}_{k,l}^T + \sum_{k,l,r,q} \tilde{\mathbf{d}}_k^T \tilde{\mathbf{d}}_r \tilde{\mathbf{F}}_l \mathbf{s}_{k,l} \mathbf{s}_{r,q}^T \tilde{\mathbf{F}}_q^T \right), \tag{12}$$

where $\tilde{\mathbf{F}}_l$ is the Toeplitz matrix of the convolutional filter $\mathbf{f}_l$. The convolutional matching pursuit for updating the spike train is summarized in Algorithm 2, and several convergence criteria can be used such as maximum number of iterations with a subsequent threshold, or the decrease of the reconstruction error between subsequent steps is less than a threshold [9].

---

**Algorithm 2** Convolutional Matching Pursuit for inferring $\mathbf{s}_{k,l}$

---

1: **Input:** Data matrix $\mathbf{X}$, number of basis functions $K$ and of temporal filters $L$, filters $\mathbf{f}_l$ and temporal descriptions $\tilde{\mathbf{D}} = [\tilde{\mathbf{d}}_1 \ldots \tilde{\mathbf{d}}_K]$.
2: **Initialization:** $\mathbf{s}_{k,l} = \mathbf{0} \; \forall k, l$, and active set $\Lambda_{t,k} = \{1, \ldots, L\} \; \forall k, t = \{1 \ldots n\}$
3: **while** (not converged) **do**
4:     **for** $k = 1, \ldots, K$ and $l = 1, \ldots, L$ **do**
5:         **Update residual gradient**: $\dfrac{\partial}{\partial \mathbf{s}_{k,l}} := \tilde{\mathbf{F}}_l^T \mathbf{X}^T \tilde{\mathbf{d}}_k - \sum_{q,r} \tilde{\mathbf{d}}_k^T \tilde{\mathbf{d}}_q \tilde{\mathbf{F}}_l^T \tilde{\mathbf{F}}_r \mathbf{s}_{q,r}$,
6:     **end for**
7:     **Find** $[c^\star, i^\star, j^\star] = \arg\max\limits_{c,i,j \in \Lambda_{c,i}} \dfrac{\partial}{\partial s_{c,i,j}}$
8:     **Set** $s_{c^\star, i^\star, j^\star} = s_{c^\star, i^\star, j^\star} + \dfrac{\partial}{\partial s_{c^\star, i^\star, j^\star}}$
9:     **Remove from active set** $\Lambda_{c^\star, i^\star} = \Lambda_{c^\star, i^\star} - \{j^\star\}$
10: **end while**
11: **Output:** Updated the firing location and activation pattern per each basis function $\mathbf{s}_{k,l}$.

---

### 3.5 Update Spatial filters H

Given the inferred locations and type of cells and its corresponding temporal description $\tilde{\mathbf{u}}_k$, we would like to learn the cell appearance of each cell type $\mathbf{H}_j$ so that minimizes the following reconstruction error:

$$\min_{\mathbf{H} \geq 0} \left\| \mathbf{X} - \sum_{k=1}^{K} \text{vec} \left( \sum_{j=1}^{J} \mathbf{D}_{k,j} * \mathbf{H}_j \right) \tilde{\mathbf{u}}_k^T \right\|_F^2 \quad \text{s.t.} \quad \|\mathbf{H}_j\|_F^2 \leq 1, \forall j, \tag{13}$$

This filter learning can be accomplished using different strategies such as a projected gradient descent, block-coordinate descent, or a closed-form formulation. The gradient descent is well-known

first-order optimization algorithm that iteratively moves towards the direction pointed by the negative gradient of the objective function. This requires to introduce a logarithm barrier function in Eq. (13) [2] in order to tackle the positivity constraints. In spite of the slow convergence rates, we found that gives poor performance on learning meaningful appearances as also mentioned in [10, 9]. Instead, block-coordinate descent has been successfully applied to learn overcomplete dictionaries [10, 6]. This technique aims to infer a spatial filter (or a dictionary element for a general dictionary learning) while the others are kept fixed, and is formulated as follows:

$$
\min_{\mathbf{H}_j \in \mathcal{H}} \left\| \mathbf{X} - \sum_k \text{vec} \left( \sum_{i \neq j} \mathbf{D}_{k,i} * \mathbf{H}_i \right) \tilde{\mathbf{u}}_k^T - \sum_k \text{vec} \left( \mathbf{D}_{k,j} * \mathbf{H}_j \right) \tilde{\mathbf{u}}_k^T \right\|_F^2
$$
$$
= \min_{\mathbf{H}_j \in \mathcal{H}} \left\| \mathbf{E}_j - \sum_k \mathbf{D}_{k,j} * \mathbf{H}_j \tilde{\mathbf{u}}_k^T \right\|_F^2 \tag{14}
$$

where $\mathbf{E}_j$ is the residual reconstruction using all the filters except for the current filter and $\mathcal{H} = \left\{ \mathbf{H} \text{ s.t. } \mathbf{H} \geq 0 \text{ and } \|\mathbf{H}\|_F^2 \leq 1 \right\}$. Eq. (14) can be minimized on a closed–form least square problem or based on a patch–based dictionary learning [10] proposed by Szlam *et al.* [11]. The latter consists of extracting all the volume patches where that $j^{th}$ cell type is active regardless of the activation pattern, and then projecting them to the spatial domain according to the corresponding activation pattern. This procedure exploits the sparsity on both space and temporal domain to collect a set of image patches that are approximated by the $j^{th}$ cell appearance $\mathbf{H}_j$ and its temporal contribution. Then, the learning of $j^{th}$ cell appearance $\mathbf{H}_j$ can be formulated as a the minimization of the reconstruction error by decomposing all the extracted image patches into two components of rank 1. This procedure is similar to patch-based dictionary learning [10, 6, 5] but we only infer a two components of rank 1 instead of an overcomplete decomposition. However, the performance of this procedure depends on the spatio-temporal sparsity of the sequence since some of these volume patches may be overlap in space and time in the whole sequences and hence they could introduce some undesired artifacts on the filters.

In contrast, the closed-form formulation handles overlapping volume patches because it considers the whole sparse representation and exploits the convolutional properties. Hence, the close-form solution of learning the $j^{th}$ cell appearance is written with some algebra as

$$
\min_{\mathbf{H}_j \in \mathcal{H}} \left\| \mathbf{E}_j - \sum_k \text{vec} \left( \mathbf{D}_{k,j} * \mathbf{H}_j \right) \tilde{\mathbf{u}}_k^T \right\|_F^2
$$
$$
= \min_{\mathbf{H}_j \in \mathcal{H}} \left\| \mathbf{E}_j - \sum_k \tilde{\mathbf{D}}_{k,j} \mathbf{H}_{:,j} \tilde{\mathbf{u}}_k^T \right\|_F^2
$$
$$
= \min_{\mathbf{H}_j \in \mathcal{H}} \left\| \mathbf{E}_j - \left[ \tilde{\mathbf{D}}_{1,j} \ldots \tilde{\mathbf{D}}_{K,j} \right] \left[ \mathbf{I}_K \otimes \mathbf{H}_{:,j} \right] \tilde{\mathbf{U}}^T \right\|_F^2
$$
$$
= \min_{\mathbf{H}_j \in \mathcal{H}} \left\| \text{vec} \left( \mathbf{E}_j \right) - \left( \tilde{\mathbf{U}} \otimes \tilde{\mathbf{D}} \right) \text{vec} \left( \mathbf{I}_K \otimes \mathbf{H}_{:,j} \right) \right\|_F^2
$$
$$
= \min_{\mathbf{H}_j \in \mathcal{H}} \left\| \text{vec} \left( \mathbf{E}_j \right) - \left( \tilde{\mathbf{U}} \otimes \tilde{\mathbf{D}} \right) \mathbf{M} \mathbf{H}_{:,j} \right\|_F^2, \tag{15}
$$

where $\tilde{\mathbf{U}} = [\tilde{\mathbf{u}}_1 \ldots \tilde{\mathbf{u}}_K]$, $\tilde{\mathbf{D}}_{k,j}$ is the Toeplitz matrix of the convolution image $\mathbf{D}_{k,j}$, $\otimes$ is the Kronecker product, $\mathbf{H}_{:,j}$ is the vectorization of the filter $\mathbf{H}_j$, $\mathbf{I}_K$ denotes the identity matrix of size $K$, and $\mathbf{M}$ is an operator matrix that maps $\mathbf{H}_{:,j}$ to $\text{vec} \left( \mathbf{I}_K \otimes \mathbf{H}_{:,j} \right)$.

Finally, Eq. (15) can be extended to all the spatial filters, and hence we can avoid this iterative update of the filters and tackle spatially and temporally overlapping among spatial filters. Therefore, the minimization w.r.t all the spatial filters is written as follows:

$$\min_{\mathbf{H}\in\mathcal{H}} \left\| \mathbf{X} - \sum_{k=1}^{K} \text{vec}\left( \sum_{j=1}^{J} \mathbf{D}_{k,j} * \mathbf{H}_j \right) \tilde{\mathbf{u}}_k^T \right\|_F^2$$

$$= \min_{\mathbf{H}\in\mathcal{H}} \left\| \mathbf{X} - \sum_{k,j} \tilde{\mathbf{D}}_{k,j} \mathbf{H}_{:,j} \tilde{\mathbf{u}}_k^T \right\|_F^2$$

$$= \min_{\mathbf{H}\in\mathcal{H}} \left\| \mathbf{X} - \sum_{k} \left( \sum_{j} \tilde{\mathbf{D}}_{k,j} \mathbf{H}_{:,j} \right) \tilde{\mathbf{u}}_k^T \right\|_F^2$$

$$= \min_{\mathbf{H}\in\mathcal{H}} \left\| \mathbf{X} - \sum_{k} [\tilde{\mathbf{D}}_{k,1} \ldots \tilde{\mathbf{D}}_{k,J}] \mathbf{H} \tilde{\mathbf{u}}_k^T \right\|_F^2$$

$$= \min_{\mathbf{H}\in\mathcal{H}} \left\| \mathbf{X} - [\tilde{\mathbf{D}}_{1,1} \ldots \tilde{\mathbf{D}}_{1,J} \ldots \tilde{\mathbf{D}}_{K,1} \ldots \tilde{\mathbf{D}}_{K,J}] \left[ \mathbf{I}_K \otimes \mathbf{H} \right] \tilde{\mathbf{U}}^T \right\|_F^2$$

$$= \min_{\mathbf{H}\in\mathcal{H}} \left\| \mathbf{X} - \tilde{\mathbf{D}} \left[ \mathbf{I}_K \otimes \mathbf{H} \right] \tilde{\mathbf{U}}^T \right\|_F^2$$

$$= \min_{\mathbf{H}\in\mathcal{H}} \left\| \text{vec}\left( \mathbf{X} \right) - \left( \tilde{\mathbf{U}} \otimes \tilde{\mathbf{D}} \right) \text{vec}\left( \mathbf{I}_K \otimes \mathbf{H} \right) \right\|_F^2$$

$$= \min_{\mathbf{H}\in\mathcal{H}} \left\| \text{vec}\left( \mathbf{X} \right) - \left( \tilde{\mathbf{U}} \otimes \tilde{\mathbf{D}} \right) \mathbf{M} \mathbf{H} \right\|_F^2, \tag{16}$$

where $\mathbf{H}$ is the column-concatenation of all the vectorized spatial filters $[\mathbf{H}_{:,1}; \ldots; \mathbf{H}_{:,J}]$ and $\mathbf{M}$ is an operator matrix that maps $\mathbf{H}$ to produce $\text{vec}\left( \mathbf{I}_K \otimes \mathbf{H} \right)$.

### 3.6 Update Temporal filters f

Given the inferred firing locations and the activation pattern for each cell $\tilde{\mathbf{d}}_k$, we would like to learn the different activation patterns $\mathbf{f}_l$ so that minimizes the following reconstruction error:

$$\min_{\mathbf{F}\geq 0} \left\| \mathbf{X} - \sum_{k=1}^{K} \tilde{\mathbf{d}}_k \left[ \sum_{l=1}^{L} \mathbf{s}_{k,l} * \mathbf{f}_l \right]^T \right\|_F^2 \quad \text{s.t.} \|\mathbf{f}_l\|_2^2 \leq 1, \forall l. \tag{17}$$

The learning of the temporal filters can be accomplished using the same procedure than the previous section (Sect. 3.5). Hence, we only update the formulas w.r.t. the activation patterns. The block-coordinate descent is written as:

$$\min_{\mathbf{F}\in\mathcal{F}} \left\| \mathbf{X} - \sum_{k} \tilde{\mathbf{d}}_k \left[ \sum_{i\neq l} \mathbf{s}_{k,l} * \mathbf{f}_l \right]^T - \sum_{k} \tilde{\mathbf{d}}_k \left[ \mathbf{s}_{k,l} * \mathbf{f}_l \right]^T \right\|_F^2 = \min_{\mathbf{F}\in\mathcal{F}} \left\| \mathbf{E}_l - \sum_{k} \tilde{\mathbf{d}}_k \mathbf{f}_l^T \tilde{\mathbf{S}}_{k,l}^T \right\|_F^2 \tag{18}$$

where $\mathbf{E}_l$ is the residual reconstruction using all the filters except for the current filter, $\mathcal{F} = \left\{ \mathbf{F} \text{ s.t. } \mathbf{f} \geq 0 \text{ and } \|\mathbf{f}\|_2^2 \leq 1 \right\}$ and $\tilde{\mathbf{S}}_{k,l}$ is the Toeplitz matrix of the convolution of the $k^{th}$ spike train of the $l^{th}$ filter, $\mathbf{s}_{k,l}$.

The close-form formulation of a single temporal filter is

$$\min_{\mathbf{f}_l \in \mathcal{F}} \left\| \mathbf{E}_l - \sum_k \tilde{\mathbf{d}}_k \mathbf{f}_l^T \tilde{\mathbf{S}}_{k,l}^T \right\|_F^2$$

$$= \min_{\mathbf{f}_l \in \mathcal{F}} \left\| \mathbf{E}_l - \left[ \tilde{\mathbf{d}}_1 \ldots \tilde{\mathbf{d}}_K \right] \left[ \mathbf{I}_K \otimes \mathbf{f}_l^T \right] \left[ \tilde{\mathbf{S}}_{1,l} \ldots \tilde{\mathbf{S}}_{K,l} \right]^T \right\|_F^2$$

$$= \min_{\mathbf{f}_l \in \mathcal{F}} \left\| \mathbf{E}_l - \tilde{\mathbf{D}} \left[ \mathbf{I}_K \otimes \mathbf{f}_l^T \right] \mathbf{S}^T \right\|_F^2$$

$$= \min_{\mathbf{f}_l \in \mathcal{F}} \left\| \text{vec} \left( \mathbf{E}_l \right) - \left( \mathbf{S} \otimes \tilde{\mathbf{D}} \right) \text{vec} \left( \mathbf{I}_K \otimes \mathbf{f}_l^T \right) \right\|_F^2$$

$$= \min_{\mathbf{f}_l \in \mathcal{F}} \left\| \text{vec} \left( \mathbf{E}_l \right) - \left( \tilde{\mathbf{S}} \otimes \tilde{\mathbf{D}} \right) \mathbf{M} \mathbf{f}_l \right\|_F^2 . \tag{19}$$

Finally, the generalization for *jointly* learning all the filters is the following:

$$\min_{\mathbf{f} \in \mathcal{F}} \left\| \mathbf{X} - \sum_{k,l} \tilde{\mathbf{d}}_k \mathbf{f}_l^T \tilde{\mathbf{S}}_{k,l}^T \right\|_F^2$$

$$= \min_{\mathbf{f} \in \mathcal{F}} \left\| \mathbf{X} - \sum_k \tilde{\mathbf{d}}_k \left( \sum_l \mathbf{f}_l^T \tilde{\mathbf{S}}_{k,l}^T \right) \right\|_F^2$$

$$= \min_{\mathbf{f} \in \mathcal{F}} \left\| \mathbf{X} - \sum_k \tilde{\mathbf{d}}_k \left[ \mathbf{f}_1^T \ldots \mathbf{f}_L^T \right] \left[ \tilde{\mathbf{S}}_{k,1} \ldots \tilde{\mathbf{S}}_{k,L} \right]^T \right\|_F^2$$

$$= \min_{\mathbf{f} \in \mathcal{F}} \left\| \mathbf{X} - \sum_k \tilde{\mathbf{d}}_k \mathbf{f}^T \tilde{\mathbf{S}}_k^T \right\|_F^2$$

$$= \min_{\mathbf{f} \in \mathcal{F}} \left\| \mathbf{X} - \left[ \tilde{\mathbf{d}}_1 \ldots \tilde{\mathbf{d}}_K \right] \left[ \mathbf{I}_K \otimes \mathbf{f}^T \right] \left[ \tilde{\mathbf{S}}_1 \ldots \tilde{\mathbf{S}}_K \right]^T \right\|_F^2$$

$$= \min_{\mathbf{f} \in \mathcal{F}} \left\| \mathbf{X} - \tilde{\mathbf{D}} \left[ \mathbf{I}_K \otimes \mathbf{f}^T \right] \mathbf{S}^T \right\|_F^2$$

$$= \min_{\mathbf{f} \in \mathcal{F}} \left\| \text{vec} \left( \mathbf{X} \right) - \left( \mathbf{S} \otimes \tilde{\mathbf{D}} \right) \text{vec} \left( \mathbf{I}_K \otimes \mathbf{f}^T \right) \right\|_F^2$$

$$= \min_{\mathbf{f} \in \mathcal{F}} \left\| \text{vec} \left( \mathbf{X} \right) - \left( \tilde{\mathbf{S}} \otimes \tilde{\mathbf{D}} \right) \mathbf{M} \mathbf{f} \right\|_F^2 . \tag{20}$$

## 4 Results

### 4.1 Artificial Sequences

Fig. 1 shows the ground truth for the synthetic data with 20 cells with two similar appearances, RA equal to 7 and two activation pattern with different decay function. Our sparse space-time deconvolution is able to identify and monitor neuronal activity at single cell while learns the cell appearance and the underlying activation patterns, and also able to discriminate among cell type given the appearance filters.

### 4.2 Computational Cost Comparison

Average computational time for each method and the proposed method with and without filter learning are shown in Table. 1.

Learning cell shapes and impulse responses is basically the bottleneck of the algorithm and the main reason of this high computational cost. Once the filters are learned, i.e. from a set of training se-

| Method | Time (sec) |
|---|---|
| Adina | 190[1] |
| NMF+Adina | 2100 |
| CSBC+SC | 60[1] |
| CSBC+STD | 100 |
| SSTD (w filter learning) | 900 |
| SSTD (w/o filter learning) | 430 |

Table 1: Comparison of computational cost

quences, the proposed method can use these a priori information and only infer the cell locations and the spike train faster than before and comparable with the other algorithms that requires to conduct all the analysis for each new sequence and without exploiting a prior information. The following table shows the average computational time for each method and also includes the proposed method with and without the learning of the filters.

### 4.3 Inferring spike trains

Fig. 2 shows that, unsurprisingly, best results are obtained when methods use the true impulse response rather than learning it from the data. Given the true impulse response, both FAST and STD fare better than Peeling, showing that a greedy algorithm is faster but gives somewhat worse results. Even when learning the impulse response, STD outperforms FAST and Peeling but is worse than using the true impulse.

### 4.4 Real Sequences

Fig. 3 and 4 show the learned cell appearances and impulse response for different number of filters and random initializations. Despite of random initialization, the proposed formulation tends to learn similar cell appearances and impulse responses.

## Footnotes

[1] parts of the code are written in c++ to speed-up the analysis

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

Figure 1: Example of ground truth for the synthetic data. From top to bottom: outline of all neurons used to generate the artificial sequence superimposed on a maximum intensity projection across the free-noise image sequence and their respective Calcium transients; the same outline but on the noisy images; and finally, the inferred cells projected image and the approximated Calcium transients by the convolution of a spike train and two activation pattern filters.

Figure 2: Receiver operating characteristic (ROC) of different methods for spike train estimation. Methods that need to learn the activation pattern perform worse than those using the true (but generally unknown) activation pattern and its parameters. FAST is at an advantage here because it infers the spike magnitude at each time step instead of a binary variable as in STD or Peeling.

Figure 3: Qualitative results of learning cell appearance and impulse response on two real data sets. The data set on the left column shows mostly cell bodies, while the data set on the right shows both cell bodies (large) and dendrites (small). For each data set, the left shows the two learned cell appearance and the right shows two learned impulse responses for multiple random initializations.

Figure 4: Qualitative results of learning cell appearance and impulse response on two real data sets. The data set on the left column shows mostly cell bodies, while the data set on the right shows both cell bodies (large) and dendrites (small). For each data set, the left shows the three learned cell appearance and the right shows the three learned impulse responses for multiple random initializations.