[Reviews · NeurIPS 2014]

Submitted by Assigned_Reviewer_1

Comments:
This paper presents a complete method for calcium imaging analysis with few user-adjustable parameters. This high-quality work is clearly written and significant to a large number of researchers working with calcium imaging data, and the chosen approach of simultaneously optimising all parameters of the analysis is original. There are a few issues that the authors should address to finalise this work.

Major issues:
1) Why is the limit on the number of types of impulse responses also K, the number of cells? More importantly, why do the authors choose only 2 impulses for the analysis? It seems quite clear from Figure 5 that the two filters are the calcium indicator responses to one or multiple action potentials. Either reducing the number of time points for impulses (i.e. reduce F) or increasing the number of possible impulses (increase L) would be wise in order to capture the range of actual impulses that are observed in the fluorescence response to the many action potentials that may occur in the 3 second long time window that is currently used.
2) One of the key advantages of the proposed method is the apparent reduction in the amount of “fiddling” necessary to go from raw calcium imaging data to observed events in cells in known positions. The authors admit “the price to pay for this elegance .. is a much higher computational cost” without going into detail. If this method is to be useful to researchers besides the authors, a detailed quantification of this “price” is needed. Can this method be run in a reasonable amount of time on a decent PC, or is a computational cluster required? How much longer does this optimisation take compared to the other methods the authors test?
3) Figure 5 is difficult to interpret. First of all, the learned “truly sparse” representation cyan and magenta symbols are difficult to see even when zooming in digitally, and they appear on both left and right panels of traces even though the legend claims they are on the right. Additionally, how are there are what appear to be signals in the right panel in locations without symbols? Lastly, this data appears to come from a paper in which cell-attached recordings were performed. If at all possible, the best possible test would be to see whether the “truly sparse” impulse times match actual recorded action potentials. Of course, the cell-attached data in the cited paper is zoomed in tightly on the recorded neuron, so the analysis problem is much reduced, but if any other data were available along these lines, that could lead to a highly calibrated method that would be used by many researchers.

Minor issues:
Please do not capitalise Calcium imaging.
Page 1 line 040 “only a fraction of neuron fires” should be “only a fraction of neurons fire”
Page 3 line 137 It was unclear until nearly the end of the paper exactly what an “impulse” or “pattern” of activity would like, perhaps the authors could plot one of the filters from the final figure earlier on or give a more precise description at the first usage of the term. The first mention of the length of these ‘patterns’ says “the number of taps or coefficients F that the learned impulse responses may have”. What is meant by “taps” ?
Page 3 line 139 “six .. parameters” I count 5 (K,H,J,L,F) unless the authors are also counting the background smoothness parameter described in the next section.
Page 5 line 258 typo “we disregards”
Page 5 line 263 please italicise priori if also italicising “a”
Page 7 line 356 typo “SSTD does as well the” should be “as well as the”
Summary: This paper presents a complete method for calcium imaging analysis with few user-adjustable parameters which, with a few additions and edits, will likely be quite useful to a large number of researchers working with calcium imaging data.

Submitted by Assigned_Reviewer_23

Summary:

The authors present a method for analyzing calcium imaging data (SSTD). This method focuses on two common analyses related to imaging data: temporal deconvolution for inferring spike trains and ROI detection / cell segmentation. The contribution of the paper is an optimization problem that solves both problems jointly.

Namely, the optimization problem identifies four features of the data: cell locations, cell shapes, spike times, and impulse responses.

In its comparison to other methods, SSTD emphasizes truly sparse representations and does not rely on heuristic pre/post-processing.

SSTD is tested against other methods and is found to have superior performance.

Quality:

The careful comparison to other methods is a strong point of the paper, particularly the results from figure 4. The block coordinate descent strategy is sound. The optimization formalism itself (Eq 1 & 2) is a parsimonious and elegant solution to the problem.

The ability to identify different cell shapes (and impulse responses) is interesting, but may also be limiting. It would be interesting to be able identify elongated regions (dendrites/axons) and separate those from cell bodies. But this might require a large bank (large J) to capture different orientations of the non-circular structures. Thus, the ability to identify different cell shapes seems limited to circular shapes, at which point it is unclear how much is actually gained by this ability.

There is little discussion on the computational complexity of this method. Further, there are no supporting analyses about computational cost vs other methods. Even a discussion would improve the quality of the paper, since it is unclear whether the improved performance of SSTD is worth the (presumed) extra cost -- this would certainly be important for very large datasets or experiments requiring online analysis.

What is the motivation for allowing different cells to have different impulse responses? Does the method find that certain neurons prefer one filter over another?

Finally, it is unclear how much is gained by joint optimization compared to solving both problems sequentially -- e.g. cell segmentation first on time-averaged data followed by sparse temporal deconvolution; here, both steps could follow the same general formalism presented in this paper. It would be interesting to know whether the performance of SSTD is mainly due to this particular formalism or if it is in fact due to joint optimization. If the latter, this could be emphasized in the paper as it would be very compelling.

Clarity:

The writing and organization of the paper are exceptionally clear.

Figure 5 is difficult to read. The spikes are nearly imperceptible. It is not clear which spike trains correspond to which calcium traces.

Originality:

The paper focuses on well-studied data analysis problems for calcium imaging. The particular approach is novel.

Significance:

This paper appears to have strong potential for widespread use in the neuroscience community.
Summary: This paper provides an elegant solution to an important calcium imaging analysis problem. The method shows superior performance compared to other methods, but the paper would benefit slightly from a more careful discussion of its potential drawbacks (computational cost, etc.)

Submitted by Assigned_Reviewer_31

The manuscript introduces a new method for joint segmentation and deconvolution of calcium imaging signals. Calcium imaging is likely to become one of the core tools of neuroscience in the coming years and any signal processing tool that can lower the level of manual drudgery is a welcome improvement. The writing is clear and describes what seems to be a useful contribution.

I just have the following remarks:
1. The authors have made an effort in comparing their method to the state-of-the-art. The results are interesting but would have been more convincing if they had evaluated segmentation performance on real data by comparing to human-segmented cells.
2. False positives in segmentation (non-cells labelled as cells) can really mess up an analysis, far worse than non-spikes labelled as spikes. Is there a way of controlling false positives? What about the risk of double counting (a single cell artificially divided by segmentation)?
3. In the optimisation program (eq. 2) the background signal is estimated non-parametrically via total variation penalisation. Why not do the same for impulse responses? Also, isn't there a risk that the background signal term starts explaining too much?

My only caveat with respect to impact is that the manuscript is more a signal processing paper than a classical NIPS paper, so I'm not sure how much it will appeal to the NIPS community. I put in a impact score of 1 but I really have no idea.

Summary: Useful contribution to the analysis of calcium imaging signals. Calcium imaging is probably the most promising experimental techniques in neuroscience.
Author Feedback
Author rebuttal: We thank the Rs for their careful comments. Reviews are mainly positive about the significance of the contribution, experiments, novelty and clarity of the paper. All the suggestions and comments made by the reviewers will be taken into account for a more complete manuscript.

@All Rs
* Computational Cost
Average computational time for each method and the proposed method with and without filter learning:

Method Time (sec)
Adina ~ 190*
NMF+Adina ~ 2100
CSBC+SC ~ 60*
SSTD (w filter learning) ~ 900
SSTD (w/o filter learning) ~ 430
*parts of the code are written in c++ to speed-up the analysis.

The learning of the cell shapes and impulse responses is the most costly step. Once the filters are learned, the proposed method can use this information and only infer the cell locations and the spike train faster than before and comparable with the other algorithms.

* Fig. 5
Fig. 5 will be improved to show the learned “truly sparse” representation in both domains, and to avoid compression artifacts that delete symbols with low intensity where there are signals.

@R1

*“Why is the limit on the number of types of impulse responses also K, the number of cells?”

We use distinct upper limits K and L for the number of cell types and impulse responses.

Such upper limits are among the most important parameters for all the matrix decomposition-based methods. They indicate the maximum number of latent signals in the sequence, and help to restrict the search space. By using two rather than one impulse response as in related work we mainly wanted to show that the latter is too coarse an approximation. We will use more than two responses in the final version.

*“Lastly, …the “truly sparse” impulse times match actual recorded action potentials”

To the best of our knowledge, this kind of data does not exist yet for all cells in large field-of-views. Unfortunately, we do not even have access to calcium image sequences plus local electrophysiological recordings as yet. We are supposed to get these over time and will conduct a more complete benchmarking in subsequent and more applied work. Here, our main goal is to demonstrate viability of a new algorithmic approach.

In view of the lack of benchmarking ground truth, we have resorted to realistic synthetic experiments. To the best of our knowledge, this paper is the first one to compare multiple spike inference algorithms at least on synthetic data, with different measures and under different noise levels and firing rates.

@R23

*“The ability to identify different cell shapes and impulse responses is interesting.....”

Learning the appearance of elongated regions is very challenging not only for the proposed method but also for all the related work. In addition, most of the proposed methods and experimental studies focus basically on detecting cell centroids. The proposed algorithm can identify randomly oriented oblique objects at the expense of using a large bank of filters. In the data used in our experiments, dendrites are orthogonal to the imaging plane and hence appear roundish.

*“What is the motivation for allowing different cells to have different impulse responses?”

There are three main motivations: 1) different cell types have different responses to the same Calcium indicators, 2) cell soma will have different response than dendrites perpendicular to the image plane, and 3) one cell can use different temporal signatures over time if that fits the data well[1*].

[1*]Reichinnek et al. Journal of Neuroscience. 2010.

*“How much is gained by joint optimization compared to solving both problems sequentially”

This is a good question. We will conduct the experiments and add them to the results section. Even now, we already show the performance of cell segmentation on time-averaged data followed by the sparse coding proposed in Adina to infer the Calcium traces.

Methods that find segmentations / ROIs from time-projected data only clearly have to operate on data with much reduced information content. For instance, cells with spurious neural activity or with low background calcium concentration are difficult to distinguish based on time-averaged data alone. In contrast, the joint optimization allows to segment cells on multiple and differently weighted time-averaged data thanks to the alternation between sparse temporal and spatial deconvolution.
@R31

*“... had evaluated segmentation performance on real data by comparing to human-segmented cells...”

This is reserved for a future more applied paper in which we need to setup a protocol for a proper evaluation as the number of annotations per sequence, how to merge the annotations, differentiation between active and silent cells, etc.

*“Is there a way of controlling false positives? What about the risk of double counting?”

The evaluation is computed frame by frame and counts a false positive in segmentation each time this “cell” is inferred as active. Doubly counted cells contribute a true and false positive. So the reported results give an honest account of the risk of these errors.

*“In the optimisation program (eq. 2) the background signal is estimated via TV”

The proposed model can be extended to include total variation smoothing of the impulse response, but we have preferred to keep the formalism simpler and make do without another parameter to tune.

*“isn't there a risk that the background signal term starts explaining too much?”

There is this risk if the corresponding parameter is chosen too small; but it did not seem to be a problem for the data used here.